# Superiority of [^11^C]methionine over [^18^F]deoxyglucose for PET Imaging of Multiple Cancer Types Due to the Methionine Addiction of Cancer

**DOI:** 10.3390/ijms24031935

**Published:** 2023-01-18

**Authors:** Yutaro Kubota, Toshihiko Sato, Chihiro Hozumi, Qinghong Han, Yusuke Aoki, Noriyuki Masaki, Koya Obara, Takuya Tsunoda, Robert M. Hoffman

**Affiliations:** 1AntiCancer Inc., 7917 Ostrow Str., San Diego, CA 92111-3604, USA; 2Department of Surgery, University of California, San Diego, CA 92037-7220, USA; 3Division of Internal Medicine, Department of Medical Oncology, Showa University School of Medicine, Tokyo 142-8666, Japan; 4Utsunomiya Central Clinic, Tochigi 321-0112, Japan; 5AntiCancer Japan Inc., Chiba 270-1505, Japan

**Keywords:** methionine, PET, methionine addiction, Hoffman effect, lung cancer, bladder cancer, rectal cancer, renal-cell carcinoma, granuloma

## Abstract

Positron emission tomography (PET) is widely used to detect cancers. The usual isotope for PET imaging of cancer is [^18^F]deoxyglucose. The premise of using [^18^F]deoxyglucose is that cancers are addicted to glucose (The Warburg effect). However, cancers are more severely addicted to methionine (The Hoffman effect). [^11^C]methionine PET (MET-PET) has been effectively used for the detection of glioblastoma and other cancers in the brain, and in comparison, MET-PET has been shown to be more sensitive and accurate than [^18^F]deoxyglucose PET (FDG-PET). However, MET-PET has been limited to cancers in the brain. The present report describes the first applications of MET-PET to cancers of multiple organs, including rectal, bladder, lung, and kidney. The results in each case show that MET-PET is superior to FDG-PET due to the methionine addiction of cancer and suggest that the broad application of MET-PET should be undertaken for cancer detection.

## 1. Introduction

Positron emission tomography (PET) was originally developed independently by Wrenn et al. [1] and Sweet et al. [2] in 1951. PET requires a type of isotope that has a short half-life and must be synthesized in a cyclotron in the same facility as the PET imager [3]. [^18^F]deoxyglucose has been the isotope of choice for most cancer PET imaging [4]. The glucose-analog isotope for PET is based on the discoveries of Otto Warburg a century ago that cancers are addicted to glucose because they metabolize glucose anaerobically through the glycolysis pathway, even under aerobic conditions [5]. This may be due in part to the hypoxic conditions of tumors. [^18^F]deoxyglucose PET (FDG-PET) has been applied to a variety of cancers. However, the glucose utilization of tumors is often not that much greater than the surrounding normal tissue, thereby limiting the effective signal of [^18^F] deoxyglucose PET.

The basis of [^11^C]methionine PET (MET-PET) is the methionine addiction of cancer [6]. The methionine dependence of cancer was first observed in 1959 when Sugimura et al. [7] observed that the removal of methionine from rat chow slowed the growth of rat tumors more than the removal of other amino acids from the rat chow. Fourteen years later, Chello and Bertino [8] found that replacing methionine with its precursor homocysteine arrested the growth of leukemia and lymphoma cells in culture. It was thought at the time that cancer cells could not synthesize methionine from homocysteine [9]. However, we showed that cancer cells make very large amounts of methionine from homocysteine [6] and require an exogenous source of methionine, indicating that cancer cells were just dependent on methionine but addicted to methionine. The methionine addiction of cancer is termed the Hoffman effect [10]. Cancer cells also seem to be addicted to folic acid and its derivatives, which are necessary for methionine biosynthesis needed for the addiction of cancer cells [11]. Cancer cells are selectively depleted of S-adenosyl-methionine (SAM) under methionine restriction, and their ratio of SAM to S-adenosyl-homocysteine (SAH) is highly reduced [12], and the levels of free methionine are depleted [13,14]. The methionine addiction of cancer cells is due, at least in part, to highly elevated trans-methylation reactions [15,16,17,18,19,20]. Thus MET-PET depends on the methionine addiction of cancer for its strong signal. Methionine addiction appears to be general to all cancers [21,22,23].

Although FDG-PET has a high sensitivity for the detection of malignant disease, it has disadvantages relating to the detection of certain cancers, such as tumors located in organs with a naturally high glycolytic metabolism (such as central-nervous-system tumors). Additionally, [^18^F] deoxyglucose cannot differentiate between inflammation and cancer since both processes are characterized by an increase in glucose metabolism. Concerning anatomy, it is difficult for FDG-PET to detect tumors close to the bladder because [^18^F]deoxyglucose is excreted by the urinary system. Furthermore, [^18^F]deoxyglucose is less sensitive in tumors of low malignancy. Based on these facts, we present here cancer cases that show the superiority of MET-PET using [^11^C]methionine, which is readily synthesized in a radiology clinic (Figure 1).

## 2. Results

### 2.1. Case 1

A 72-year-old female diagnosed with Stage IIIA rectal cancer underwent resection in 2015. After surgery, the patient received adjuvant chemotherapy (capecitabine) for six months. However, liver metastasis developed in 2016, and lung metastasis developed in 2018 and 2019. Since every recurrence was isolated, the patient underwent surgery for each recurrence. In January 2020, computed tomography detected multiple mediastinal lymph node nodules, and FDG-PET showed an accumulation of [^18^F]deoxyglucose in these nodules (Figure 2A). Hence, systemic chemotherapy was planned. However, MET-PET showed no accumulation in these lesions (Figure 2B), which were thereby diagnosed as granulomas or inflammatory lymph-node swelling. Since then, the patient has been carefully observed, but no increase in lymph-node swelling was observed. In this case, toxic chemotherapy was avoided since MET-PET distinguished inflammatory lymph-node swelling from lymph-node metastasis.

### 2.2. Case 2

A 69-year-old male diagnosed with bladder cancer received transurethral resection of a bladder tumor (TURBT), followed by radiotherapy (60 Gr) in 2017. Two years later, multiple lung tumors were detected on both sides of the lung apex. FDG-PET showed the accumulation of [^18^F]deoxyglucose in both lesions (Figure 3A). However, MET-PET imaging showed the accumulation of [^11^C]methionin only on the right side of the lung apex (Figure 3B). From these results, it was determined that only the lesion on the right side of the lung apex is a cancer recurrence. After that, a transbronchial lung biopsy (TBLB) was performed and confirmed that the right-side lesion was a cancer recurrence and the left lesion was a granuloma. Since it was a single recurrence, stereotactic body radiation therapy (SBRT) was performed.

### 2.3. Case 3

A 76-year-old female presented with a small nodule in her lower right lung at an annual health checkup in 2018. This nodule gradually increased for two years. Therefore, FDG-PET and MET-PET were performed in 2020. MET-PET showed accumulation, but FDG-PET did not (Figure 4A,B). This result indicated that this lesion was well-differentiated adenocarcinoma. Even without a pathological diagnosis, this combination of FDG-PET and MET-PET examination can diagnose the malignancy of a lesion.

### 2.4. Case 4

A 55-year-old female was originally diagnosed with a parotid gland tumor and a solitary lung metastasis. The primary tumor was resected, and the patient received radiotherapy for lung metastasis. She was subsequently diagnosed with renal cell carcinoma. CT examination showed renal cell carcinoma and a lower-right-lobe lung nodule. FDG-PET showed no accumulation in the lung nodule (Figure 5A), but MET-PET showed accumulation in the lung nodule (Figure 5B). A biopsy was performed, and the pathological diagnosis was renal cell carcinoma. Thus MET-PET indicated a biopsy even when FDG-PET did not show accumulation.

## 3. Discussion

In brain cancers, such as glioblastoma or metastatic tumors, the usefulness of MET-PET has been shown [24,25]. Because the brain is a naturally high glycolytic-metabolism organ, the FDG-PET of the tumor has a low signal, and there is difficulty in distinguishing the normal brain from cancer. MET-PET is currently used in some radiology clinics to determine the radiotherapy efficacy on brain cancers. Nettelbladt reported a comparison between FDG-PET and MET-PET for non-small cell lung cancer patients [26]. In this report, the sensitivity was 93%, the specificity was 100%, and the accuracy was 95% for MET-PET, 87%, 75%, and 84%, respectively, for FDG-PET. Although the number of cases is small, this report indicates that MET-PET has higher accuracy than FDG-PET for cancers other than in the brain. The present report shows the superiority of MET-PET over FDG-PET for multiple cancer types.

FDG-PET cannot discriminate between inflammation and cancer because both are characterized by an increase in glucose metabolism. Inflammation and cancer are sometimes difficult to distinguish. This causes misreading of the staging or the choice of treatment in oncology. In contrast, since methionine does not accumulate in inflammation, MET-PET can distinguish between inflammation and cancer.

It is difficult for FDG-PET to diagnose cancer when the tumor exists close to the bladder because [^18^F]deoxyglucose is excreted by the urinary system. Therefore, the superiority of MET-PET is also shown, especially in cancers in the pelvic cavity, such as urologic cancer [27], prostate cancer [28], and uterine cancer [29].

In addition, even low-grade malignant lesions demonstrate enhanced [^11^C]methionine accumulation due to the methionine addiction of cancer in contrast to glucose uptake [28,30]. MET-PET should be effective in the diagnosis of tumors of low-grade malignancy that cannot be diagnosed using FDG-PET. There are reports that have shown the difficulty of diagnosing well-differentiated adenocarcinoma of the lung [31] or invasive lobular carcinoma (ILC) of the breast using FDG-PET [32]. In contrast, MET-PET had efficacy in diagnosing low-grade tumors, such as low-grade glioma [30] and low-Gleason-score prostate cancer [28]. In case 3, MET-PET detected well-differentiated adenocarcinoma of the lung without biopsy.

## 4. Materials and Methods

All of the patients fasted for at least 6 h before the MET-PET examination. The patients were intravenously injected with [^11^C]methionine at a dose of 370 MBq/kg body weight and hydrated with 0.9% sodium chloride. Physical activity was kept to a minimum, with a rest period of 10 min post-injection. At 10 min after injection, PET was performed using a PET/computed tomography system (Vereos Digital PET/CT; Philips, Amsterdam, The Netherlands) CT unit and a PET scanner with 23,040 lutetium-yttrium oxyorthosilicate (LYSO) crystals in 64 rings. After that, the patients were intravenously injected with [^18^F]deoxyglucose at a dose of 4.4 MBq/kg body weight, and PET was performed the same as above.

A low-amperage CT scan was acquired for the attenuation correction of the PET images (213 mA, 120 kV, and CT slice thickness of 5 mm). The CT dose index for low-dose CT was 10.7 mGy. After non-enhanced CT, a total-body PET examination in the caudocranial direction from the upper thighs to the vertex was performed (2 min per bed). Reconstruction was performed using the 3D reconstruction method of ordered subset expectation maximization with 30 subsets and two iterations.

The images were reviewed by a nuclear physician. The standard uptake values (SUV) of the target lesions were measured.

## 5. Conclusions

Based on the above, we propose the following diagnostic procedure based on the results of both MET-PET and FDG-PET (Table 1). If both FDG-PET and MET-PET show accumulation in the tumor, it indicates high-grade malignancy. In contrast, only the accumulation of [^11^C]methionine indicates low-grade malignancy. Only the accumulation of [^18^F]deoxyglucose indicates inflammation or granuloma. If there is no accumulation in both cases, it is likely that the tumor is not malignant, although the possibility of a low-grade tumor remains.

## Figures and Tables

**Figure 1 ijms-24-01935-f001:**
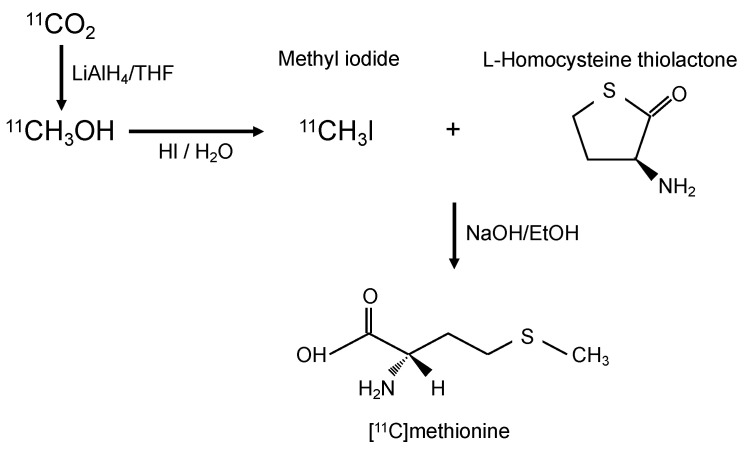
Diagram of synthesis of [^11^C]methionine. LiAlH_4_ = Lithium aluminum hydride, THF = Tetrahydrofuran, HI = Hydroiodic acid.

**Figure 2 ijms-24-01935-f002:**
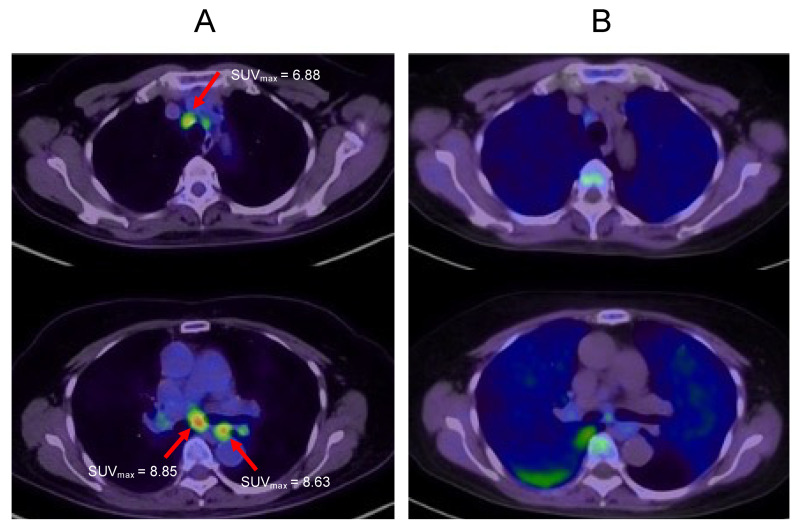
Multiple mediastinal lymph node nodules. (**A**): FDG-PET imaging. (**B**): MET-PET imaging. Arrows indicate mediastinal lymph nodes.

**Figure 3 ijms-24-01935-f003:**
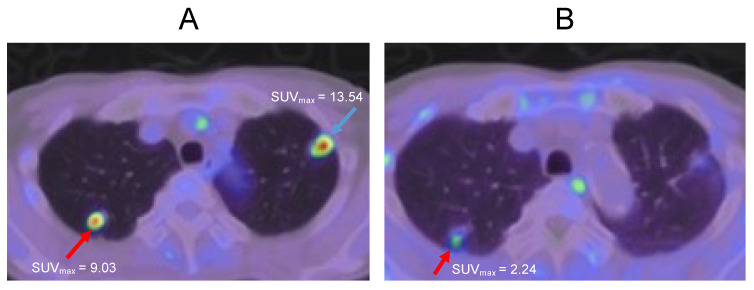
Lung tumors at both sides of the lung apex. (**A**): FDG-PET imaging. (**B**): MET-PET imaging. The red arrows indicate cancer, and the blue arrow indicates granuloma.

**Figure 4 ijms-24-01935-f004:**
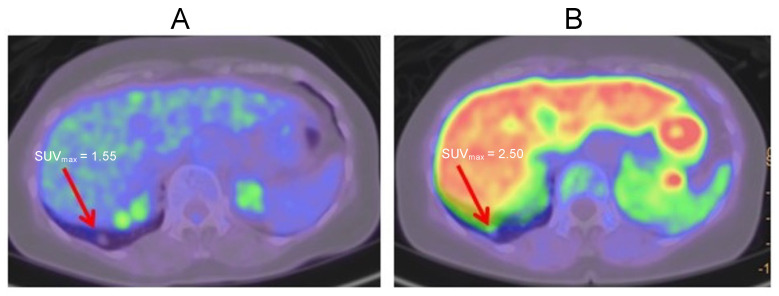
Solitary lung nodule at lower right lung. (**A**): FDG-PET imaging. (**B**): MET-PET imaging. Arrows indicate well-differentiated adenocarcinoma.

**Figure 5 ijms-24-01935-f005:**
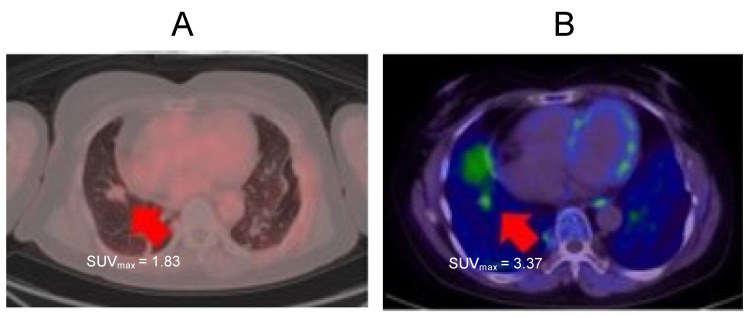
Lung nodule. (**A**): FDG-PET imaging. (**B**): MET-PET imaging. Arrows indicate lung metastasis from renal cell carcinoma.

**Table 1 ijms-24-01935-t001:** Algorism of for cancer diagnosis combining FDG-PET and MET-PET.

	FDG-PET Positive	FDG-PET Negative
MET-PET positive	High-grade malignancy	Low-grade malignancy
MET-PET negative	Inflammation/Granuloma	Benign lesion (Low-grade malignancy)

## Data Availability

The data presented in this study are available on request from the corresponding author. The data are not publicly available due to privacy.

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
