# Peer review of "Superiority of [^11^C]methionine over [^18^F]deoxyglucose for PET Imaging of Multiple Cancer Types Due to the Methionine Addiction of Cancer"

_ijms, 2023, doi:10.3390/ijms24031935_

Round 1

Reviewer 1 Report

This is well written manuscript on the important topic.

The methodology is sound, data properly collected and adequately discussed. The experimental documentation is impressive. 

There is high novelty in the presentation and the findings are significant and will advance our knowledge on human cancer.

The findings are scientifically sound and of strong interest to the general readership.

I do not have any major critique, except minor corrections in formatting references. For example, names of the authors of reference 1 should not be capitalized.

Reviewer 3 Report

In this manuscript by Kubota et al., the authors described the extended application of MET-PET ([11C] METhionine - Positron Emission Tomography) in the detection of multiple types of cancers related to different organs such as lung, colon, bladder…etc. Previously, the use of MET-PET was limited to cancers of the central nervous system (especially glioblastoma).  For this, they investigated the five different cases for different cancers and compared the traditional FDG-PET ([18F] Deoxy Glucose-PET) with MET-PET and proposes that analysis of both scans may lead to a better understanding of malignancy. The authors also discussed how inflammation detected as cancer by FDG-PET scan is a serious pitfall of the technique and MET-PET scan might be important to rule it out.

The number of cases observed here for different cancers is very small to prove the efficacy of the MET-PET scan, however, the results are very interesting.

I have the following comments for the authors' consideration-

Major Comments:

1.     The authors should describe the full protocol for both C-11 MET-PET and FDG-PET scans and analysis of the scans in a separate section of “Materials and Methods”. The authors should also discuss the mean and/ or maximum SUV (Standardized uptake value) for all the cases they have described.

Minor Comments-

1.     The authors should include additional information like which of the PET scan was performed first in the patients, and how much the time difference between both the PET scans is.

Round 2
